# A Long-Term Water Quality Prediction Method Based on the Temporal Convolutional Network in Smart Mariculture

Yuexin Fu, Zhuhua Hu *, Yaochi Zhao and Mengxing Huang *

State Key Laboratory of Marine Resource Utilization in South China Sea, School of Information and Communication Engineering, School of Computer Science and Cyberspace Security,  Hainan University, Haikou 570228, China; fuyuexin@hainanu.edu.cn (Y.F.); zhyc@hainanu.edu.cn (Y.Z.)
* Correspondence: eagler_hu@hainanu.edu.cn (Z.H.); huangmx09@163.com (M.H.)

**Abstract:** In smart mariculture, traditional methods are not only difficult to adapt to the complex, dynamic and changeable environment in open waters, but also have many problems, such as poor accuracy, high time complexity and poor long-term prediction. To solve these deficiencies, a new water quality prediction method based on TCN (temporal convolutional network) is proposed to predict dissolved oxygen, water temperature, and pH. The TCN prediction network can extract time series features and in-depth data features by introducing dilated causal convolution, and has a good effect of long-term prediction. At the same time, it is predicted that the network can process time series data in parallel, which greatly improves the time throughput of the model. Firstly, we arrange the 23,000 sets of water quality data collected in the cages according to time. Secondly, we use the Pearson correlation coefficient method to analyze the correlation information between water quality parameters. Finally, a long-term prediction model of water quality parameters based on a time domain convolutional network is constructed by using prior information and pre-processed water quality data. Experimental results show that long-term prediction method based on TCN has higher accuracy and less time complexity, compared with RNN (recurrent neural network), SRU (simple recurrent unit), BI-SRU (bi-directional simple recurrent unit), GRU (gated recurrent unit) and LSTM (long short-term memory). The prediction accuracy can reach up to 91.91%. The time costs of training model and prediction are reduced by an average of 64.92% and 7.24%, respectively.

**Keywords:** aquaculture water quality prediction; TCN deep learning; smart mariculture

## 1. Introduction

In traditional mariculture, farmers can only rely on breeding experience to control the water quality, and it is impossible to immediately grasp a drop in aquatic production caused by water quality problems. In mariculture, the timely and accurate prediction of water quality can help fishermen take countermeasures before the water environment is seriously deteriorated. The prediction of water quality can help farmers foresee future changes in water quality parameters and promptly remind farmers to take measures that make sure that the fish are living and growing in the most suitable environment. It can improve the quality of aquatic products while increasing the yield [1–3].

In terms of water quality prediction, traditional prediction algorithms mainly include the time series method [4], Markov chain method [5], regression analysis method [6,7], grey theory method [8], and support vector machine [9]. They need a small amount of historical data, but these prediction methods also have many shortcomings, such as low prediction accuracy, and do not consider many factors, such as air temperature. Compared with traditional algorithms, artificial neural networks and deep learning models have higher robustness and accuracy, and adapt to multiple complex environments. Liu et al. [10] used the BP (back propagation) neural network to build a multi-scale dissolved oxygen nonlinear prediction model. Zhang et al. [11] and Eze et al. [12] used the LSTM (long short-term memory) model to predict water parameters in the long and short term, but

the long-term prediction effect was poor. Han et al. [13] proposed a radial basis function neural network neural network with a flexible structure by which a model was established to predict water data. Ye et al. [14] used multi-task and multi-view learning methods to fuse multiple data sets and combined spatial correlation to predict urban water quality. However, urban waters are not as complex and changeable as sea water, and they consider fewer influential factors. Liu et al. [15,16] used the SRU (simple recirculation unit) network and improved deep Bi-SRU (bi-directional stacked simple recirculation unit) network to predict multiple water quality parameters. However, these methods still have problems, such as a complex network structure and high time complexity, which increase the hardware requirements. The spatial cross-correlation analysis method [17] combined with spatial distribution information can solve the inaccuracy and volatility of correlation analysis, especially if the sample is insufficient. Because of a large number of data samples in this paper, we use the Pearson correlation coefficient method [18] to analyze the direct correlation of different water quality factors.

In order to solve the problems above, we combine Pearson's correlation coefficient analysis to obtain the prior information between the water quality data, and build and train a deep learning model based on the TCN (temporal convolutional network). The TCN is a variant of a convolutional neural network that handles sequence modeling tasks. The convolution structure of TCN is simple and has high-efficiency and long-term memory ability. This paper uses the pre-trained TCN model to predict the dissolved oxygen, pH, and water temperature data. Finally, the prediction model is evaluated from a multi-faceted perspective.

Our main contributions can be summarized as follows:

- The relationship and influencing factors between water quality parameters are analyzed. The Pearson correlation coefficient method is used to obtain prior information among dissolved oxygen, water temperature, pH and other water quality parameters, which can be used as the input parameters of the model training.
- A water quality prediction model based on the TCN network is constructed and trained. Compared with the RNN, LSTM, GRU (gated recurrent unit), SRU and BI-SRU prediction models, the TCN prediction model not only improves the accuracy of prediction but also requires less time complexity, especially in training and prediction.

The rest of this paper is organized as follows. Section 2 gives the methods of data acquisition and data correlation analysis. Section 3 presents the water quality prediction model based on TCN. In Section 4, the experimental results are analyzed and discussed, the accuracy of the model is evaluated, and the time complexity of the prediction model is analyzed. Finally, Section 5 summarizes this paper and outlooks future work.

## 2. Water Quality Data Acquisition and Analysis

### 2.1. Data Acquisition

In this paper, we use real data for research. The data were collected at the sea aquaculture base in Xincun Township, Lingshui County, Hainan Province, China. Data collection was realized by deploying sensor equipment in the cage. The collected data were stored in the data server, and real-time data information could be viewed on the mobile terminal. The sensor and equipment for water quality data collection are shown in Figure 1.

Because long-distance wireless data transmission is prone to noise interference, and the equipment is susceptible to seawater corrosion and individual data is missing, it is necessary to preprocess the collected data. Firstly, this paper uses the linear difference method and threshold processing to fill in the missing information, and then uses smoothing filtering and wavelet transform to filter out the noise interference in the water quality data, aiming to correct and correct the abnormal water quality data. After the above-mentioned series of processing, we clarified the water quality data and provided a good foundation for the follow-up forecasting process.

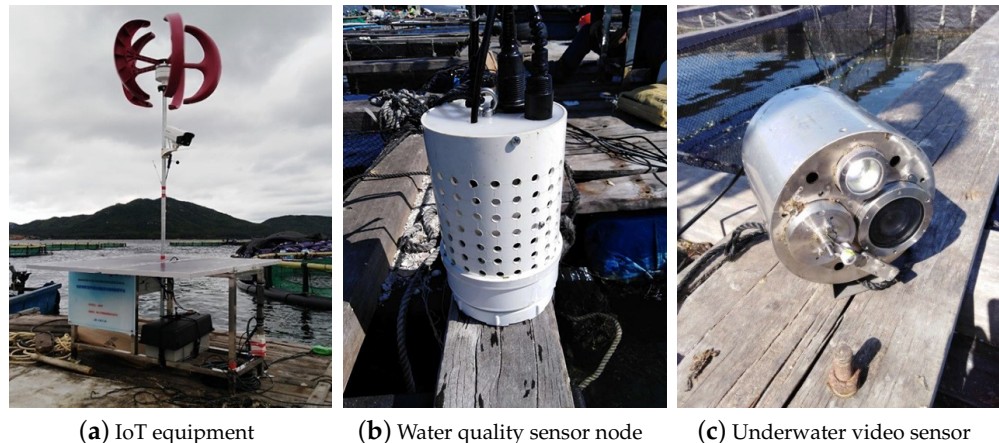

(**a**) IoT equipment      (**b**) Water quality sensor node      (**c**) Underwater video sensor

**Figure 1.** Experimental data acquisition site. (**a**) IoT equipment transmitted data to mobile phone and farmers can monitor water quality in real time. (**b**) Water quality sensor node collected water quality factor data such as water temperature, pH and dissolved oxygen. (**c**) Underwater video sensor collected and recorded underwater video data.

We visited fishermen and professional farmers, and learned that fishermen need to feed about 5–6 times a day, and the number of times will change due to weather changes. Cold weather is relatively less frequent than feeding bait. At the same time, continuous precipitation weather will have related effects on fish culture. Regardless of whether it is feeding bait or precipitation, the data of our sensor are always being recorded.

Through the dynamic relationship between the various water quality parameters, the various water quality parameters influence and reflect each other. For example, precipitation will have a chain reaction effect on water temperature, pH and other parameters, and adding feed will slightly affect the pH and salinity. However, when the external environment is raining, our system records the changes of multiple water quality parameters and displays them in the training dataset. Finally, we use the deep learning network to automatically detect the changing trends of various parameters under different precipitation conditions. The network model digs out the relationship between the hidden parameters. As a nonlinear adaptive dynamic system, the neural network has the advantage of extracting the internal characteristics of information through self-learning, and it is very suitable to explore the hidden relationship between data.The basic characteristic values of the predicted water quality factors are shown in Table 1.

**Table 1.** The basic characteristic values of the predicted water quality factors.

| | Water Temperature (°C) | pH | DO (mg/L) | Salinity (PSU) | Precipitation (mm) | Air Temperature (°C) |
|---|---|---|---|---|---|---|
| Max | 28.59 | 8.73 | 8.66 | 36.44 | 0.01143 | 31.26 |
| Min | 20.70 | 7.99 | 0.63 | 31.65 | 0 | 15.97 |
| Median | 25.69 | 8.28 | 4.58 | 34.67 | 0 | 22.45 |
| Average | 25.50 | 8.29 | 4.51 | 34.59 | 0.000395 | 22.82 |

*2.2. Correlation Analysis*

Before establishing the key parameter in the water quality prediction model, this paper uses Pearson's correlation coefficient method [18] to analyze the correlation of key water quality parameters, such as water temperature, pH, dissolved oxygen, and salinity. The Pearson correlation coefficient method is commonly used to qualitatively measure and analyze the strength of the correlation degree between variables. It is defined as the quotient of the covariance and the standard deviation between two variables. We calculate the Pearson correlation coefficient between the water temperature, dissolved oxygen, pH

and other factors which can influence water quality. The calculation results are shown in Table 2.

**Table 2.** Result of correlation analysist.

| Parameters | Water Temperature | pH | DO | Salinity | Precipitation | Air Temperature |
|---|---|---|---|---|---|---|
| Water temperature | 1.0 | 0.3228606 | −0.35295023 | −0.35055917 | −0.18294075 | 0.36917955 |
| pH | 0.3228606 | 1.0 | 0.41250716 | −0.27341231 | −0.07272953 | 0.20661753 |
| DO | −0.35295023 | 0.41250716 | 1.0 | −0.07251022 | 0.14161098 | 0.03189754 |

It can be seen from the above Table 2 that the water temperature has a negative correlation with salinity and solubility, a positive correlation with air temperature, and a weaker negative correlation with pH and precipitation. At the same time, pH has a strong positive correlation with dissolved oxygen, a moderate positive correlation with water temperature and air temperature, a medium negative correlation with salinity, and basically no correlation with precipitation. In addition, dissolved oxygen shows a strong positive correlation with pH, a negative correlation with water temperature, and almost no correlation with salinity and air temperature.

## 3. The Prediction Model Based on TCN Deep Learning

### 3.1. Temporal Convolutional Network

The temporal convolutional network is based on the convolutional neural network and introduces dilated convolution, causal convolution and residual modules. It can be used as an alternative to the recurrent neural network (RNN) to better deal with time series prediction. The overall flowchart of TCN network is shown in Figure 2. In time series forecasting, the LSTM network solves the problem of RNN network gradient explosion, but the memory gate of LSTM will gradually forget part of the historical information [19,20], and all historical information cannot be used. The temporal convolutional network can effectively combine historical data for long-term prediction through the dilation causal convolution and residual module. TCN networks are causal in each layer structure, which means that "leakage" of future information or historical data will not occur [21,22]. Improved SRU (simple recurrent unit—SRU) [23], and GRU [24] networks have improved model efficiency and prediction effects, but there are also problems in processing variable length sequences. TCN can obtain the same output sequence length regardless of the input sequence length, which increases the flexibility of the model. At the same time, TCN can process time series data in parallel, which greatly reduces the time cost of the model. The time domain convolutional network is based on the convolutional neural network and introduces the expansion convolution, causal convolution and residual modules. It can be used as an alternative to the RNN (recurrent neural network) to better deal with the time series prediction problem.

We define the time series sequence of the input water quality parameters prediction model as $x_0, \ldots, x_T$ and the goal of the model prediction is to output prediction values $\hat{y}_0, \ldots, \hat{y}_T$ for a period in the future based on the observed historical data. Our prediction model is to obtain the function $\mathcal{F}$ and make it produce the following map:

$$\hat{y}_0, \ldots, \hat{y}_T = \mathcal{F}(x_0, \ldots, x_T) \tag{1}$$

The process of prediction model training is to continuously optimize the function to minimize the loss between the real value and the predicted value.

Causal convolution [25]: To prevent the occurrence of information "leakage", TCN adopts causal convolution, that is, the convolution operation is performed in the order of time series. For example, the convolution output at time $t$ is only related to the data at time $t$ and $t-1$ or earlier. The TCN network adopts the form of a one-dimensional fully connected network, and uses zero-padded operations to make the length of the hidden layer and the input layer always consistent.

$$(F^*X)(x_t) = \sum_{k=1}^{K} f_k x_{t-(K-k)} \tag{2}$$

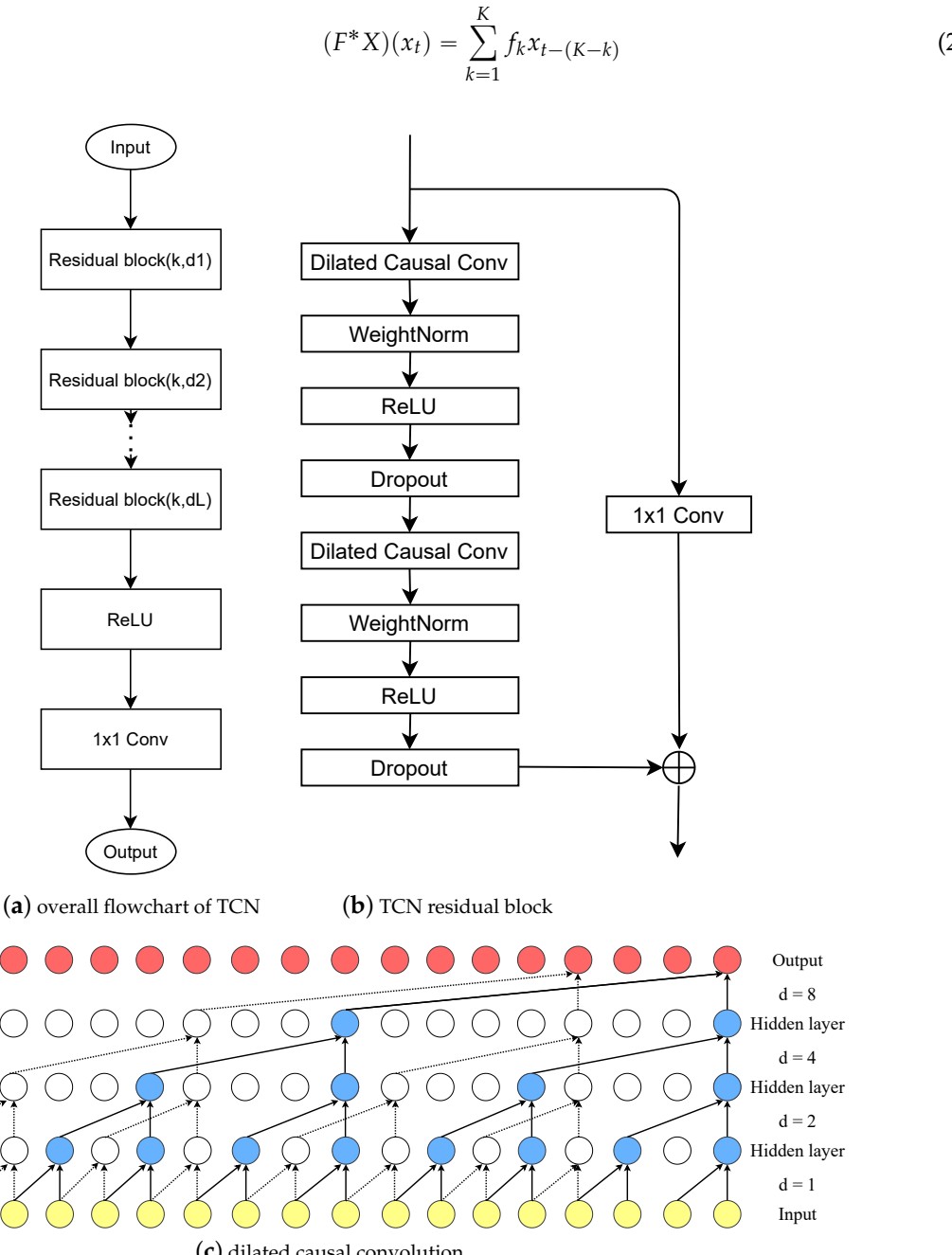

(**a**) overall flowchart of TCN  (**b**) TCN residual block

(**c**) dilated causal convolution

**Figure 2.** Construction of TCN. (**a**) Overall flowchart of TCN. (**b**) TCN residual block. The residual block is stacked in two layers, and each layer is composed of dilated causal convolution and nonlinear layers. (**c**) A dilated causal convolution with dilation factors $d = [1, 2, 4, 8]$ and filter size $k = 2$. The receptive field expands with the increase of d and k which means that more of the input sequence is covered.

Dilated convolution [26]: Dilated convolution is defined as follows:

$$\left(F_d^*X\right)(x_t) = \sum_{k=1}^{K} f_k x_{t-(K-k)d} \tag{3}$$

When $d = 1$ in the above equation, it means ordinary convolution. In the process of water quality prediction, the field of view of the convolutional layer can be expanded by increasing the number of convolution kernels $k$ or expanding the expansion factor $d$ in

order to dig deeper into the historical water quality data so that the model can learn longer time series data. To achieve this goal, in actual situations, the expansion factor $d$ is usually expanded exponentially with a base of 2. Figure 2c shows dilation causal convolution, with $d = [1, 2, 4, 8]$ and $k = 2$ as an example.

The residual block [27] is represented by $f$, which performs a series of transformations on the input $x$, and then adds it to $x$ for output operations.

$$o = \text{ReLU}(x + f(x)) \tag{4}$$

For the deeper network depth $n$, the larger $k$ and d become very important in TCN since the receiving range of TCN depends on the network depth $n$, the filter size $k$ and expansion coefficient $d$. Each layer contains multiple filters for feature extraction. Therefore, in the design of the general TCN model, we use the general residual module to replace the convolutional layer.

The composition of the residual module is shown in Figure 2b. From Figure 2, we can see that the residual block is stacked in two layers, and each layer is composed of dilated causal convolution and nonlinear layers. After the expanded convolution, spatial dropout is added for regularization. In addition, the input and output have different lengths in the TCN network, and the residual input $x$ and $f(x)$ are added directly. Therefore, the $1 * 1$ convolution operation is introduced between the identity mapping of the residual block in order to ensure the same tensor scale.

### 3.2. Evaluation Metrics

In the above training model, we introduced three evaluation metrics [28] to evaluate the prediction effect, which are defined as follows:

MAE (mean absolute error): MAE is the basic evaluation metric, and the following methods are generally used as a reference to compare the advantages and disadvantages.

$$\text{MAE} = \frac{1}{N} \sum_{i=1}^{N} |y_i - \hat{y}_i| \tag{5}$$

RMSE (root mean squared error): RMSE denotes the mean error, which is more sensitive to extreme values. If there is an extreme value in the training process at some time points, RMSE is greatly affected by the increasing error. The change in the evaluation index can be used as the benchmark for the robustness test of the model.

$$\text{RMSE} = \sqrt{\frac{1}{N} \sum_{i=1}^{N} (|y_i - \hat{y}_i|)^2} \tag{6}$$

MAPE (mean absolute percent error): MAPE considers not only the deviation between the predicted data and the real data, but also the ratio between the deviation and the real data.

$$\text{MAPE} = \frac{1}{N} \sum_{i=1}^{N} \frac{|y_i - \hat{y}_i|}{y_i} \tag{7}$$

### 3.3. Construction of Prediction Model Based on TCN

The overall process of the water quality parameters prediction model based on TCN is shown in Figure 3. Firstly, the water quality parameters collected by various sensors is received through the wireless network; then, the missing water quality data are filled by the linear difference method and the average filtering and wavelet transform are used to correct the abnormal water quality data and remove noise. Secondly, the Pearson correlation coefficient analysis method is used to analyze the correlation among water quality factors, such as water temperature, dissolved oxygen, and pH, to obtain relevant prior information. Finally, the pre-processed water quality factors and the relevant prior information obtained

from the analysis are input into the constructed model for training. When the error of the trained model meets the requirements, we save the model for subsequent testing and otherwise readjust the hyperparameters and retrain the model.

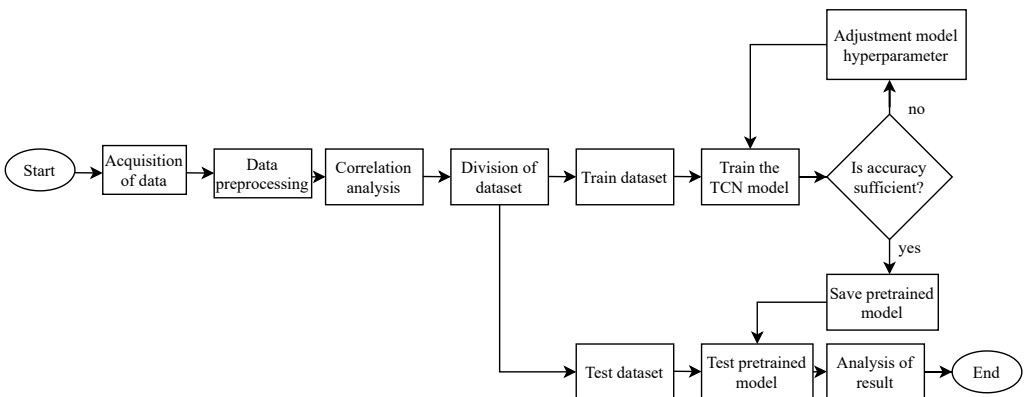

**Figure 3.** Flowchart of water quality prediction model.

The model in this article is based on the Keras framework and the Python programming language. In addition to the TCN model, the RNN, LSTM, GRU, SRU and BI-SRU prediction models are also built for comparison in experiments. The analysis of the correlation coefficients of the various water quality parameters above is used as the prior information, and at the same time, 20,000 sets of water quality parameters are input into the model for training. In order to control the variables to better compare the prediction effect, the input dimension of each model is 6 and the output dimension is 1, and each model is trained for 50 epochs. The batch size is set to 64 after comprehensively considering the training time and convergence speed. Especially in the TCN prediction model, the size of the convolution kernel (kernel size) $k$ in each convolution layer is 4, and the expansion coefficient $d$ is $[1, 2, 4, 8, 16, 32]$. The description of water quality prediction model is shown in Algorithm 1.

---

**Algorithm 1:** Description of water quality prediction model.

---

　　**Data:** $X = (x_0, \ldots, x_T)$, $d = [1, 2, 4, \ldots, L]$ and hyperparameter
　　**Result:** prediction value $\hat{Y} = \hat{y}_0, \ldots, \hat{y}_T$
**1** Fill the missing and correct abnormal data;
**2** Analyze the correlation degree between key water parameter;
**3** Initialize network weights and thresholds;
**4** **while** *stop condition is not met* **do**
**5** 　　**for** $d = 1; d \leqslant L; d = d * 2$ **do**
**6** 　　　　**for** $i = 0; i \leqslant 1; i = i + 1$ **do**
**7** 　　　　　　Dilated causal convolution for X: $F_d^* X$;
**8** 　　　　　　Weightnorm and dropout is added for regularization.;
**9** 　　　　**end**
**10** 　　　　Residual block output: $o = ReLU(x + f(x))$ ;
**11** 　　**end**
**12** **end**
**13** Save pretrained model and analysis result;

---

The trend of loss function at each epoch during training is shown below. From Figure 4, we can see that the error between the real data and the predicted data is constantly decreasing, and finally approaches zero infinitely as the training process progresses. In the early stage of the training process, the reduction is huge, and it stabilizes in the later stage. It can also be seen from Figure 4 that the TCN model has the fastest convergence speed during the training process, followed by the GRU model, and LSTM is slightly slower. At

the same time, the LSTM model will oscillate slightly after the training epoch to 20 times. This is because the loss function is in the end and the best point cannot be further reduced.

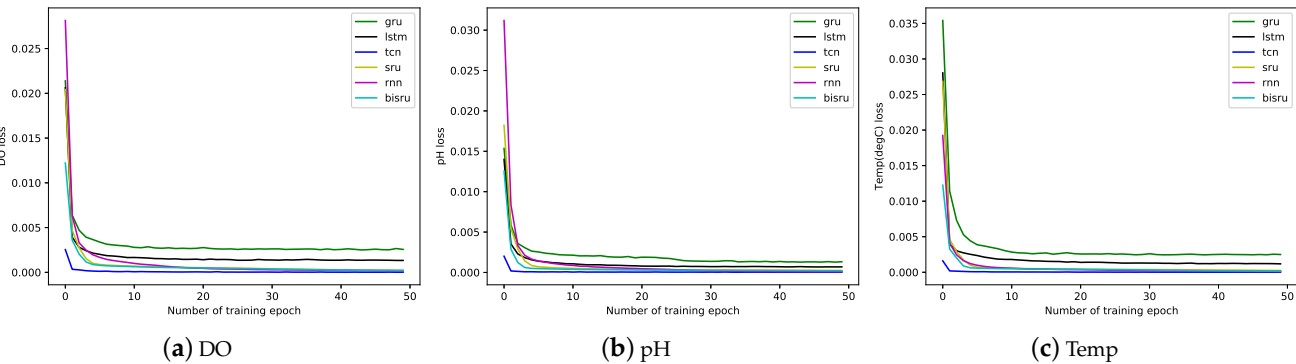

**Figure 4.** Comparison of changes in loss function of different models during model training: (**a**) dissolved oxygen, (**b**) pH, (**c**) water temperature.

## 4. Experimental Results and Discussion

The experimental data are collected from marine aquaculture cages with sensor equipment, and then transmitted to a data server for storage through a wireless bridge. The data collection interval is 5 min, including water temperature, salinity, pH and dissolved oxygen parameters. A total of 20,000 sets of data are used as experimental data for model training, and another 3000 sets of data are used for comparison of the prediction results.

The experimental environment used in this paper is a Intel(R) Core(TM) i7-7700HQ CPU @2.80GHz, 16GB RAM, NVIDIA GeForce GTX 1050Ti 4GB, Windows 10 (64-bit) operating system, minitconda3 IDE, and the construction of the network is based on the Keras 2.0 framework and Python 3.6.

The accuracy and measurement range of the sensor are shown in Table 3. F.S. is the abbreviation of "Fulle Scale" and PSU is the abbreviation of "Practical Salinity Unit".

**Table 3.** The accuracy and range ability of the sensors.

| Technical Parameters | The Type of Sensors | | | | | |
|---|---|---|---|---|---|---|
| of Sensors | Salinity | Chlorophyll | Turbidity | Water Temperature | pH | Dissolved Oxygen |
| Range ability | 0~100 PSU | 0~400 ug/L | 0.1~1000 NTU | −10~60 °C | 0~14 | 0~20.00 mg/L |
| accuracy | ±1.5% F.S. | ±3% F.S. | ±1.0 NTU | ±0.2 °C | ±0.01 | ±1.5% F.S. |

### 4.1. Comparison of Prediction Effects

We use six different neural network models, including RNN, SRU, BI-SRU, LSTM, GRU and TCN, to compare the changing trends and values of water quality parameters. According to the actual situation and the above analysis, these models respectively predict the future 3000 values of pH, water temperature, and dissolved oxygen. The sampling interval of water quality data is every five minutes. The comparison between the true value and the predicted value is shown in Figure 5 below.

It can be seen from Figure 5 that the predicted data are close to the true values, but the TCN model is better for fitting the real data trend when predicting peak and trough values. At the same time, it can be seen from Table 4 that the values of MAE, RMSE and MAPE of the TCN model are much smaller among the test dataset. Therefore, the TCN prediction model has a better prediction result on water quality parameters.

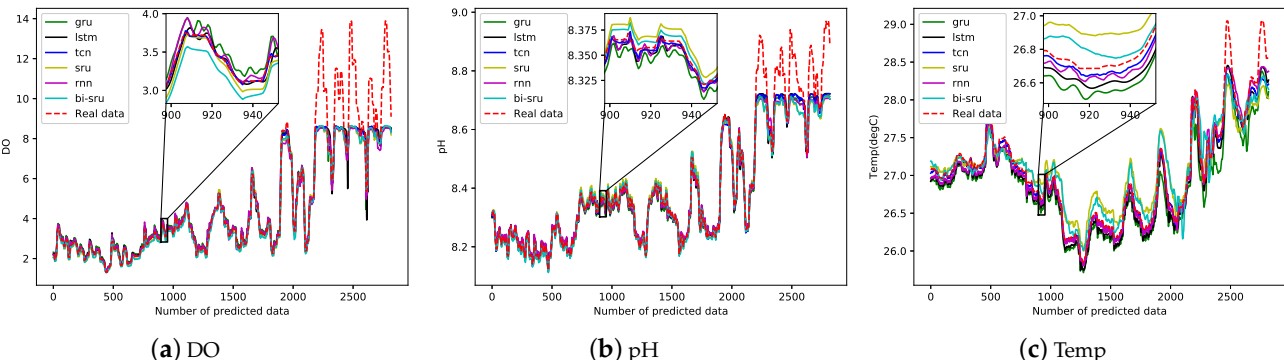

**(a)** DO　　　　　　　**(b)** pH　　　　　　　**(c)** Temp

**Figure 5.** Comparison of real and predicted values in different prediction models: (**a**) dissolved oxygen, (**b**) pH, (**c**) water temperature.

**Table 4.** Records of MAE, RMSE and MAPE in long-term prediction.

| | Water Temperature | | | pH | | | Dissolved Oxygen | | |
|---|---|---|---|---|---|---|---|---|---|
| | **MAE** | **RMSE** | **MAPE** | **MAE** | **RMSE** | **MAPE** | **MAE** | **RMSE** | **MAPE** |
| RNN | 0.2105 | 0.3185 | 0.7635 | 0.0713 | 0.1199 | 0.8210 | 1.3971 | 0.7150 | 10.8982 |
| LSTM | 0.1275 | 0.2392 | 0.4606 | 0.0299 | 0.0614 | 0.3420 | 0.4904 | 1.1672 | 5.7004 |
| GRU | 0.1750 | 0.1104 | 0.4010 | 0.0338 | 0.0678 | 0.3870 | 0.5121 | 1.2032 | 6.1280 |
| SRU | 0.1646 | 0.2204 | 0.6037 | 0.0290 | 0.0592 | 0.3325 | 0.5731 | 1.1617 | 8.5907 |
| BI-SRU | 0.1697 | 0.2276 | 0.6238 | 0.0307 | 0.0599 | 0.3527 | 0.5763 | 1.1577 | 8.8024 |
| TCN | 0.0416 | 0.0906 | 0.1152 | 0.0214 | 0.0505 | 0.2729 | 0.4014 | 1.1046 | 3.7750 |

### 4.2. Comparison of Time Cost for Training and Predition

In the experiment, we record the time required for each epoch of the six network models during the training process. The three water quality parameters of dissolved oxygen, water temperature, and pH are trained for 50 epochs. The time cost results during training are shown in the Figure 6 below. In the training process of six water quality parameter prediction models, the time spent by the TCN model is the lowest among models. This is because the recurrent neural network must predict the sequence strictly in the order of time, which means that it must wait until the prediction of the previous time step is completed before the subsequent prediction. However, the TCN model uses the same filter for each layer, and the convolution operation can be completed in parallel. Therefore, when encountering a long time series, the TCN model treats it, as a whole, in parallel, which greatly reduces the training and prediction time.

In the dissolved oxygen prediction model shown in Figure 6a, the LSTM prediction model is trained for 50 epochs and takes a total of 1511.91 s; the BI-SRU takes 1297.73 s; the GRU takes 1138.38 s; the SRU takes 672.70 s; the RNN takes 392.93 s; and the TCN takes 276.57 s. In the pH prediction model showed Figure 6b, the LSTM prediction model is trained for 50 epochs and takes a total of 1531.36 s; the BI-SRU takes 1256.78 s; the GRU takes 1204.81 s; the SRU takes 650.32 s; the RNN takes 393.80 s; and the TCN takes 275.69 s. In the water temperature prediction model shown in Figure 6c, the LSTM prediction model is trained for 50 epochs and takes a total of 1519.70 s; the BI-SRU takes 1310.67 s; the GRU takes 1185.45 s; the SRU takes 679.09 s; the RNN takes 387.70 s; and the TCN takes 277.98 s. In training, the TCN model saves 64.92% on time, on average.

After obtaining the trained model, we use the pretrained model to predict 3000 sets of future water quality values. In the dissolved oxygen prediction process, the LSTM model takes 3.24 s; the GRU model takes 2.94 s; the SRU model takes 2.28 s; the RNN model takes 2.19 s; the BI-SRU model takes 3.35 s; and the TCN takes 2.49 s. In the pH prediction process, the LSTM model takes 3.32 s; the GRU model takes 3.01 s; the SRU model takes 2.25 s; the RNN model takes 2.03 s; the BI-SRU model takes 3.74 s; and the TCN takes 2.45 s.

In the water temperature prediction process, the LSTM model takes 3.29 s; the GRU model takes 2.82 s; the SRU model takes 2.33 s; the RNN model takes 2.23 s; the BI-SRU model takes 3.39 s; and the TCN takes 2.49 s. In prediction, the TCN model saves 7.24% on time, on average.

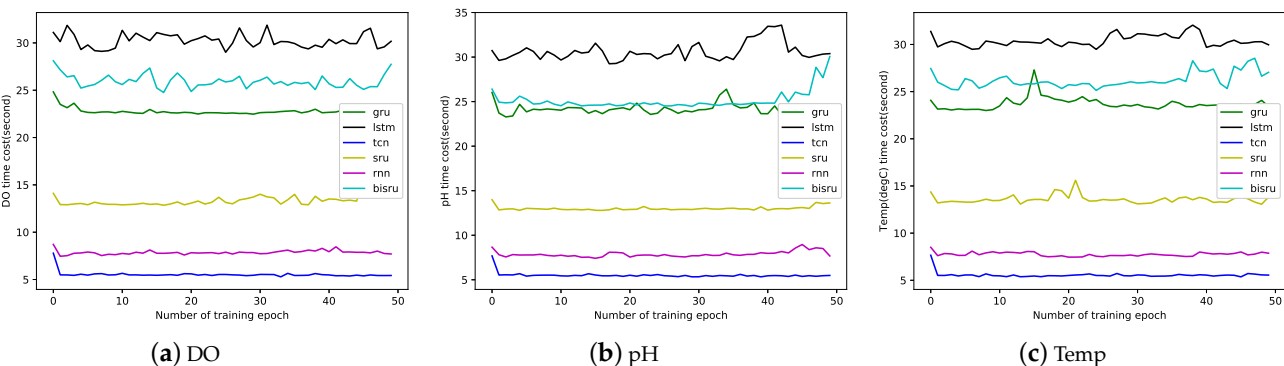

**Figure 6.** Comparison of training time under different models: (**a**) dissolved oxygen, (**b**) pH, (**c**) water temperature.

### 4.3. Discussion

From Figure 5 and Table 4, we can see that, compared with other models, the TCN water quality prediction model has higher prediction accuracy, and the difference between the the predicted value and the actual value is small. First of all, because the TCN prediction model adopts the residual network structure to solve the gradient attenuation problem, it has the ability to explore the characteristics of the deep network so that the model has a strong fitting ability and generalization ability. Secondly, the TCN model adopts the structure of dilated causal convolution, and its receptive field expands with the increase in filter size k and dilation factors d, which can cover more time series, and then can extract more historical water quality information and improve the accuracy of model predictions. Finally, the TCN model has a different back propagation path, compared to the recurrent neural network, and the gradient is more stable during the training process; it is unlikely to face the problem of gradient disappearance or gradient explosion.

From the experimental results in Section 4.2, it can be seen that the training and prediction time required for the prediction model proposed in this paper are lower than other prediction models. First of all, because the TCN prediction model uses a parallel approach, it does not need to wait for the previous moment to complete the prediction before the next moment, like a recurrent neural network. Secondly, the memory space occupied by the TCN model during the training process is also smaller. TCN filters share parameters among the same layers, and the amount of memory used only depends on the depth of the network. LSTM, GRU and SRU models need to store a large number of parameters of input gates, memory gates, and output gates during the training process. Therefore, the water quality prediction model we propose can greatly reduce the time complexity and provide fishermen with early warning information in a more timely manner. This provides a reliable and effective basis for the high-density aquaculture industry to formulate efficient water quality protection policies and measures.

Based on the above discussions, we know that our prediction method can provide fish farmers with the changing trend of water quality factors for the next 10 days. Moreover, through our method, it also can provide decision support for scientific management of aquaculture water quality. Fish farmers can check the water quality for the next 10 days at any time and take timely measures to avoid large-scale fish deaths caused by the sudden deterioration of water quality.

## 5. Conclusions and Future Work

At present, most smart aquaculture systems mainly use sensor-based IoT systems to monitor water quality in real time. This method of real-time monitoring has hysteresis. Obviously, if the water quality can be accurately predicted for a long period of time in the future, farmers can be allowed to take measures in advance, which can effectively resist breeding risks and improve production efficiency. However, the current trend of water quality is mainly based on the long-term accumulated experience of farmers for speculation. This method has the characteristics of strong subjectivity, poor reliability and timeliness.

Aiming at the long-term prediction of key water quality parameters in smart mariculture, this paper proposes a prediction method based on temporal convolutional network. After preprocessing the data with linear filling and sliding filtering, we also use the Pearson correlation coefficient to analyze the correlation of water quality parameters. Finally, we input prior information and preprocessed data into the constructed model for training. Experimental results show that, compared with RNN, SRU, BI -SRU, LSTM and GRU, the model proposed in this paper has higher prediction accuracy and lower time complexity. Specifically, this paper proposes that the long-term prediction method of water quality parameters has a prediction accuracy of 91.91%, an average reduction in training time of 64.92%, and an average reduction in prediction time of 7.24%. Therefore, in smart aquaculture, the solution we propose can meet the requirements for accurate prediction of water quality parameters values.

However, the prediction model we propose is not accurate enough for longer-term peak prediction. In the future, we will focus on the optimization of the deep learning network structure and integrate the modules related to peak prediction. In addition, in order to make the water quality prediction model more robust and further reduce the impact of climate change on the model prediction, we will collect years of water quality data as the training dataset in future studies. At the same time, we will incorporate multiple factors, such as seasonal changes and climate change, into the deep neural network as prior information so that the prediction model can achieve longer-term prediction results.

**Author Contributions:** Conceptualization, Y.F. and Z.H.; methodology, Y.F. and Z.H.; software, Y.F.; validation, Y.F.; formal analysis, Y.F.; investigation, Y.F., Z.H. and Y.Z.; resources, Z.H.; data curation, Y.F., Z.H. and M.H.; writing—original draft preparation, Y.F.; writing—review and editing, Z.H., Y.Z. and M.H.; visualization, Y.F.; supervision, Z.H.; project administration, Z.H. and Y.Z.; funding acquisition, Z.H. All authors have read and agreed to the published version of the manuscript.

**Funding:** This research was funded by the Hainan Province Natural Science Foundation of China (Grant No. 619QN195 and Grant No. 620RC564), the National Natural Science Foundation of China (Grant No. 61963012 and Grant No. 62161010).

**Conflicts of Interest:** The authors declare no conflict of interest.

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
