# Peer review of "A Long-Term Water Quality Prediction Method Based on the Temporal Convolutional Network in Smart Mariculture"

_water, doi:10.3390/w13202907_

Round 1

Reviewer 1 Report

The submitted paper is very well processed. it is its formal side.

However, in my opinion, it has several logical shortcomings. 6 parameters were selected for analysis first and then only three. Statistical analyzes were then processed for the obtained data.

With this type of article, I can't decide whether they are mainly about statistics or really about water - its quality. A correlation is created for the relationship between water temperature and DO. However, the maximum DO content at a certain water temperature is precisely physically determined in a verified table. This is not about correlation but physics. Is this simple table of oxygen content at different temperatures included in the statistical processing? Then the boundary conditions are not described. Is the solution and values ​​obtained from a closed reservois or open mariculture? If it is sea aquaculture, it is necessary to take into account the solution of mixing water with the environment and balancing concentrations, and then also the species of fish and their total weight are crucial for mixing and changing concentrations due to the movement of fish. The basic characteristic values ​​of the solved quantities should also be stated. E.g. min, max, median and the like. E.g. for Precipitation there is a difference if 500 mm of rain or 2000 mm of rain is added to the aquaculture. Do we know what pH rain water has? Is the value of salina dependent only on the properties of the water or is it influenced by the regular addition of feed? What salinity then has the feed? Therefore, I believe that this prediction method is not objective and the monitored three parameters need to be evaluated more objectively and physically, not only statistically. Therefore, I recommend adding clear information to the paper. 

Reviewer 2 Report

The paper aims to present water quality prediction method. Although it may be an interesting topic, I find many issues that concerns me.

What were the criteria for selecting the parameters? The water temperature is hardly an indicator of water quality. Why chlorophyll data were omitted?

One may disagree that ‘urban waters are not as complex and changeable as sea water, and they consider fewer Influential factors’. Urban water is usually influenced by human pressure, much larger than sea water. This is in addition to natural factors.

Reliability of the data is rising my concern. Authors use algorithms to fill in the gaps in the data chain (without reviling how many). Then the ‘produced’ data undergo yet another algorithm to predict their value in the future.

As presented in Table 1, parameters are codependent.  This fact disqualify selected parameters from the proper scientific analyses. It is commonly known that e.g. pH shows diurnal fluctuation, which results from air (and to lesser extend water) temperature.

Please, explain better Figure 2, especially part c, when there is no legend provided.

Table 2 presents accuracy of sensors which were not included in the analysis.

20000 sets of data may be impressed, but when data collection interval is 5 minutes it is not impresses, whatsoever.  They represents only two months.

The discussion section lacks discussion. Authors presents and explain results without any reference to the other findings. Discussion with published literature is required.

How is the presented method better than Markov chain method, regression analysis method, grey theory method, support vector machine?? Nothing supports this conclusion.

Authors state that the advantage of the presented method is including temperature into the analysis. However, parameter is only used in the correlation matrix, which in my opinion is redundant.  

Finally, contrary to the title, there are no long-term predictions in the manuscript.

Round 2

Reviewer 2 Report

The paper still lacks disscussion, which goal is t"o interpret and describe the significance of Authors findings in light of what was already known about the research problem being investigated and to explain any new understanding or insights that emerged as a result of your study of the problem". Untill proper discussion is provide I do not support publication.

Round 3

Reviewer 2 Report

The paper is acceptable in its present form